



# Airborne wind energy system test bench electrical emulator

Carolina Nicolás-Martín [1], David Santos-Martín [1], Francisco DeLosRíos-Navarrete [2,3], and Jorge González-García [2]

[1]Department of Electrical Engineering, Universidad Carlos III de Madrid, Avenida de la Universidad 30, 28911 Leganés (Madrid), Spain
[2]Department of Aerospace Engineering, Universidad Carlos III de Madrid, Avenida de la Universidad 30, 28911 Leganés (Madrid), Spain
[3]CT Ingenieros A.A.I. S.L. Avenida Leonardo Da Vinci 22, 28050 Getafe (Madrid), Spain
**Correspondence:** Carolina Nicolás-Martín  (canicola@ing.uc3m.es)

**Abstract.**

Airborne Wind Energy Systems (AWES) offer a promising alternative to conventional wind turbines, but their commercialization is hindered by challenges in efficiently converting the highly dynamic mechanical power of tethered flight into stable electrical energy. While extensive research has focused on optimizing AWES flight trajectories and control strategies, 5 the power conversion stage, which is critical for integrating AWES into electrical grids, is relatively under-researched. To bridge this gap, reliable and flexible electrical test bench emulators are needed to replicate AWES dynamics under controlled conditions, enabling systematic evaluation and optimization of power electronics and control strategies.

This paper presents a validated electrical test bench emulator and a torque ripple-optimized Model Predictive Control (MPC) strategy designed to enhance the performance of AWES ground station generators. The proposed emulator accurately repro-10 duces the mechanical-electrical interactions of a real AWES by simulating the variable tether forces and reeling dynamics encountered during optimal crosswind flight. Two electrical topologies are introduced: a separated DC bus configuration that closely mimics real AWES energy storage dynamics and a common DC bus topology that minimizes battery requirements for extended control testing. The proposed MPC strategy ensures precise generator speed and torque regulation, achieving less than 1% root mean square error (RMSE) in torque tracking while optimizing energy efficiency.

Using experimental flight data, the test bench demonstrates an overall energy efficiency exceeding 80% and peak conversion efficiencies up to 93%, with Permanent Magnet Synchronous Generators (PMSG) outperforming Induction Machines (IM) by 2–6% in instantaneous efficiency. These findings establish electrical test benches as essential for AWES development, offering a scalable platform for optimizing power conversion and control, which advances AWES as a viable renewable energy technology.

## 1 Introduction

Airborne Wind Energy Systems (AWES) are an innovative renewable energy technology designed to harvest high-altitude winds using autonomous, tethered aircraft. These systems offer access to stronger and more consistent wind resources than those available at lower altitudes, leading to improved capacity factors and higher energy yields (Bechtle et al., 2019). Addi-





tionally, unlike traditional horizontal-axis wind turbines, AWES eliminate the need for large, static support structures, reducing
construction costs and minimizing environmental impacts (Hagen et al., 2023).

Since their initial conceptualization in the 1980s (Loyd, 1980), AWES have undergone significant development. Over the
past two decades, researchers have proposed a variety of system architectures, ranging from small-scale prototypes (Fagiano
et al., 2014; Zgraggen et al., 2016; Wood et al., 2017; Fagiano et al., 2018; Schmidt et al., 2020; Castro-Fernández et al.,
2023) to pre-commercial systems with rated power capacities of up to 200 kW (Kitepower, 2023; SkySails Power, 2023).
These systems are commonly classified based on the location of energy conversion: onboard the aircraft via wind turbines or
on the ground using the traction forces exerted on the tether to drive a generator. Ground-generation AWES, which are the
focus of this study, typically operate in a pumping cycle consisting of two phases: the traction (reel-out) phase, where energy
is generated as the aircraft flies crosswind, and the retraction (reel-in) phase, where the aircraft is reeled back at an angle that
ensures minimal energy consumption.

Over the past decades, AWES have witnessed remarkable advancements in aerodynamic design and control, with numerous
studies optimizing tethered aircraft trajectories, lift-to-drag ratios, and energy harvesting strategies (Fagiano et al., 2022). These
developments have enabled AWES to reach high operational efficiency, making them a promising alternative to traditional wind
energy technologies. However, while aerodynamic aspects of AWES have matured significantly, the electrical power conversion
systems required for ground-generation remain relatively underexplored.

AWES operate in a highly dynamic environment where mechanical power generation fluctuates due to variations in wind
conditions, flight trajectories, and tether forces (Freeman et al., 2021). To effectively harness and stabilize this intermittent
energy, robust electrical systems and advanced control strategies are crucial. Despite extensive optimization of mechanical
power extraction, research on power conversion in AWES ground stations remains limited. As AWES move toward commer-
cialization, addressing challenges such as energy storage optimization, fault tolerance, and reactive power control is essential
to ensure reliable and scalable operation.

Existing studies (Pavković et al., 2018; Uppal et al., 2021) have made notable contributions by proposing power conversion
topologies and control strategies, including optimal damping (Pavković et al., 2018) and cascade control approaches for induc-
tion generators (Uppal et al., 2021). Nevertheless, many of these approaches would benefit from experimental validation that
incorporates AWES-specific flight data. For larger-scale systems, Coleman et al. (2014) proposed a multi-machine AWES park
using permanent magnet synchronous generators, with later studies (Ebrahimi Salari et al., 2018; Salari et al., 2019) exploring
a direct AC bus connection for offshore applications. While reducing reliance on converters, this approach raises challenges
in reactive power and tether torque control. Very advanced power converter control techniques have been explored by Magdy
Gamal Eldeeb (2019) and Saberi and Rezaie (2022) for AWES applications. Both strategies showing an enhancement on
control performance, although validation under AWES-specific conditions is still needed. Research on machine selection and
energy storage has identified electrically excited synchronous machines and permanent magnet synchronous machines with
high-energy magnets as promising candidates (Urbanek et al., 2019). Studies on power electronics (Bagaber et al., 2020) and
energy storage (Joshi et al., 2022; Pavković et al., 2014) highlight battery storage as a viable option for power smoothing,
considering its efficiency and cost-effectiveness compared to alternatives.





Significant further research is required to complement these valuable first contributions and experimental validation is key to refining AWES power conversion. Electrical test bench emulators offer a crucial tool in bridging the gap between aerodynamic advancements and electrical system maturity. By accurately replicating AWES flight conditions in a controlled environment, these emulators allow for in-depth analysis, optimization, and validation of power conversion architectures before large-scale deployment. They enable researchers to evaluate system efficiency, investigate new control methodologies, and assess the impact of various energy storage and grid-integration strategies. Previous work has contributed significantly to the development of laboratory-scale test bench emulators for AWES. For instance, Kumar et al. (2023) introduced a real-time emulator utilizing a Permanent Magnet Synchronous Generator (PMSG) with field-oriented control techniques, providing valuable insights into the emulation of airborne wind turbine dynamics. While this study advances emulator-based AWES research, further exploration is needed to assess alternative control strategies tailored to the specific challenges of AWES applications. Additionally, incorporating energy storage dynamics and alternative machine topologies could enhance the versatility and applicability of test bench emulators. Addressing these aspects, along with control approaches aimed at minimizing torque ripple, would contribute to more comprehensive and high-fidelity AWES emulation.

In response to these challenges, the study detailed in this article develops and validates a scalable, fully electrical test bench emulator for AWES ground-generation systems. The emulator is designed to replicate the dynamic behavior of real-world AWES during both reel-in and reel-out phases. Two battery connection topologies are explored: one with separate DC buses for the generating and emulating sides, and another with a common DC bus to facilitate efficient energy recirculation. A Model Predictive Control (MPC) scheme with a ripple optimization strategy is employed to ensure precise control and minimize torque ripple. Using experimental tether force and length profiles (Schmehl, 2023), the emulator is evaluated for its control performance, energy storage efficiency, and applicability to long-term dynamic testing.

The remainder of this paper is structured as follows: Section 2 presents the proposed emulator topology and control strategies. Section 3 describes the validation methodology using measured AWES flight data. Section 4 discusses the results and their implications for AWES power conversion systems. Finally, Sect. 5 concludes with key findings and recommendations for future work.

## 2 Methodology

This section outlines the methodology used to develop and validate the proposed AWES test bench emulator. The emulator is designed to replicate the dynamic behavior of a real AWES during the reel-in and reel-out phases. It begins by describing the AWES electrical power conversion system and emulator structure, detailing the interaction between key components such as the generator, emulator machine, power converters, and energy storage system. Next, the aerodynamic system inputs for AWES power conversion are introduced, defining the key flight and mechanical parameters that influence emulator operation. This is followed by a discussion of two electrical topologies for energy conversion and storage, evaluating their suitability for different testing scenarios. Finally, the model predictive control strategy is presented, including its ripple optimization approach and system discretization process.





## 2.1 AWES electrical power conversion system and emulator structure

Figure 1 illustrates the power conversion process in an AWES, control components are represented in color green for clarity. To optimize mechanical power extraction in an AWES, an electrical machine (G) regulates the kite's reeling speed, $v_{tether}$, via a

95   DC-AC power converter, ensuring consistent rotation, despite varying tether forces. During the reel-out phase, G operates as a generator, converting mechanical to electrical energy, which is transferred to a high-voltage DC bus. During the reel-in phase, G operates as a motor, spinning in the opposite direction and allowing the tether to retract. A bi-directional DC-DC converter stabilizes the bus voltage by charging a storage device (e.g., a battery) during the reel-out phase and discharging it during the reel-in phase when G functions as a motor to retract the kite. This process ensures efficient energy management and stable

100   system operation during both phases of the AWES cycle.

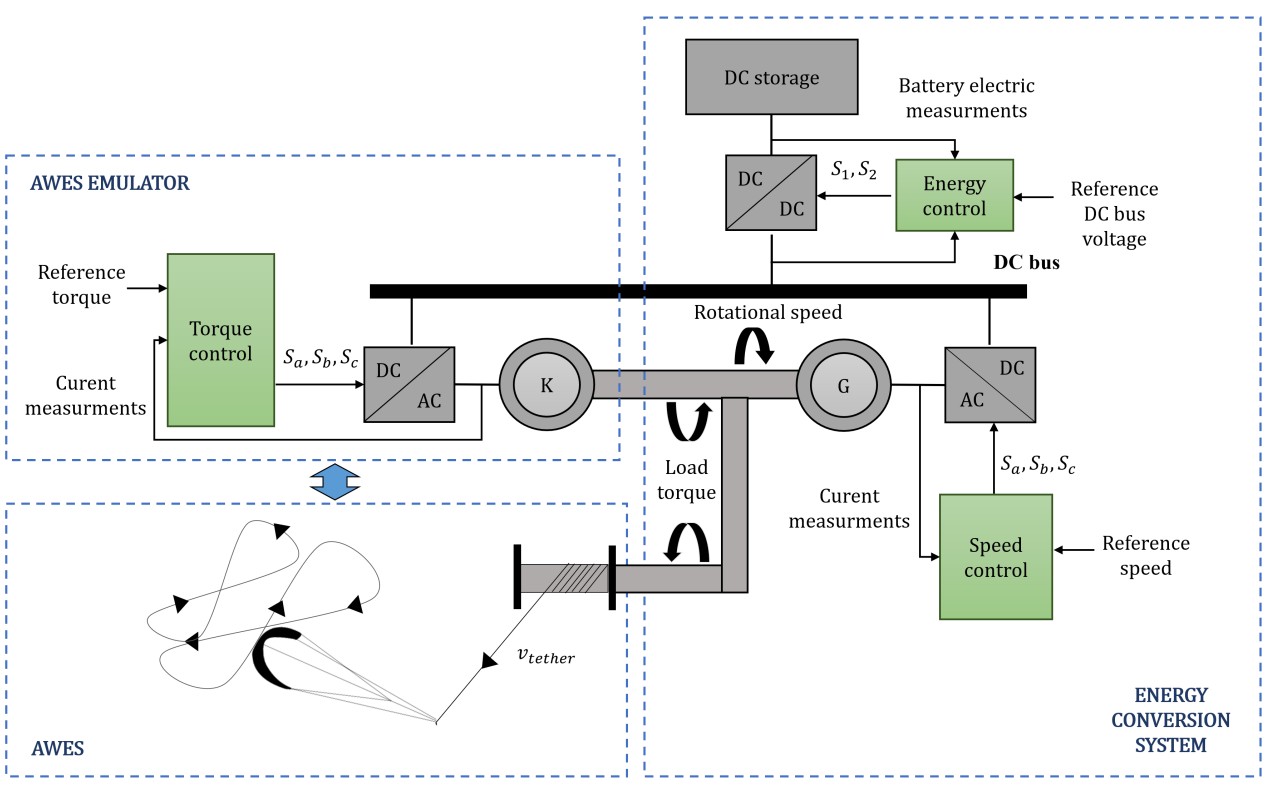

**Figure 1.** Schematic representation of the power conversion process in AWES, including the AWES emulator, energy conversion system, and control interactions.

The proposed AWES emulator simulates the effect of the changing tether forces on the electric machine shaft using another electrical machine (K) that regulates the shaft load torque via a DC-AC power converter. Precise and robust control of this converter is essential to accurately replicate the kite's mechanical behavior.



## 2.2 Aerodynamic inputs for AWES power conversion

To maximize mechanical power output during the traction phase of the cycle, an AWES aircraft must follow a closed trajectory in crosswind conditions (Loyd, 1980). Commonly used flight paths include circular and lemniscate (figure-eight) patterns, with the latter offering more consistent tether forces, $F_{\text{tether}}$, which are advantageous for power generation (Erhard and Strauch, 2015) and for grid integration (Eijkelhof et al., 2024). Along this path, tether forces increase during downward turns and decrease during climbing segments due to variations in apparent wind velocity caused by gravity.

The efficiency of mechanical power generation depends on the reeling factor $f$, which is defined as

$$f = \frac{v_{\text{tether}}}{v_{\text{wind}}}, \tag{1}$$

where $v_{\text{tether}}$ is the tether reel-out speed and $v_{\text{wind}}$ is the wind speed at the wing. The optimal reeling factor, $f_{\text{opt}}$, for straight tethers is given by (Schmehl et al., 2013)

$$f_{\text{opt}} = \frac{1}{3} \sin\theta\cos\phi, \tag{2}$$

where $\theta$ and $\phi$ are the azimuth and elevation angles of the kite respectively. These variables are used to calculate the reference torque, $T_{\text{load}}$, and rotational speed, $\omega_m$, for the emulator, which are defined as

$$T_{\text{load}} = F_{\text{tether}} \cdot R_{\text{drum}} \cdot i, \quad \omega_m = \frac{v_{\text{tether}}}{R_{\text{drum}} \cdot i}, \quad i = \frac{\omega_{drum}}{\omega_m} \tag{3}$$

where $R_{\text{drum}}$ is the drum radius and $i$ is the overall gear ratio of the system's drivetrain. For the sake of simplicity, $R_{\text{drum}}$ is assumed constant and a value of $i = 1$ is considered for this study.

To better understand how these variables influence the operation of the electrical power conversion system and kite emulator, Fig. 2 illustrates the distribution of time spent at each combination of angular speed and torque values ($T_{\text{load}}$, $\omega_m$) during a reference AWES cycle from Schmehl (2023). The reel-out phase, during which the tether is extended and mechanical power is converted into electrical energy, is characterized by a wide range of operating points. Conversely, in the reel-in phase, where the tether is retracted and energy is consumed to pull the kite back, the torque values exhibit lower variability, and the electrical machine operates as a motor rather than a generator.

## 2.3 Electric topology for an airborne wind energy system and its test bench emulator

The electric topology is fundamental for replicating the energy conversion dynamics of an AWES and enabling efficient power flow in the proposed test bench emulator. This section describes the topology for a real AWES and the two emulator configurations designed to simulate its behavior.

### 2.3.1 Topology for an airborne wind energy system

As shown in Fig. 3a, the proposed topology for a real AWES uses a kite tether wound around a drum connected to a three-phase electrical machine (G). During the reel-out phase, machine G acts as a generator, converting the kite's mechanical energy into electrical energy. During the reel-in phase, it operates as a motor, consuming energy to retract the tether.



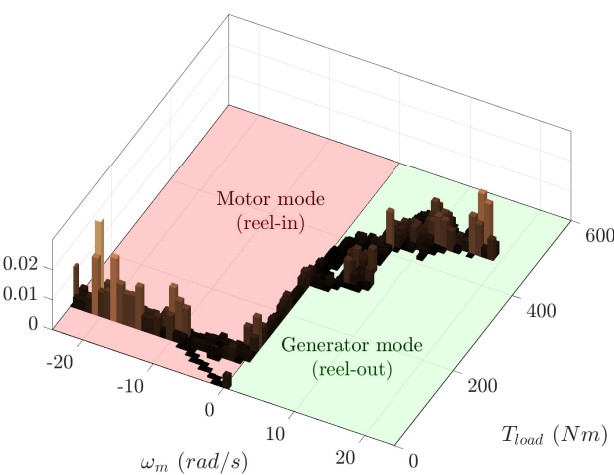

**Figure 2.** Distribution of time spent at each combination of angular speed and torque during the AWES cycle, highlighting the operational differences between the reel-out (generation) and reel-in (motor operation) phases.

Machine G is controlled by a DC-AC converter (G-C-AC) to manage power flow, while the generated energy is stored in a battery (G-B) via a DC-DC converter (G-C-DC). The system's power generation and consumption are governed by the kite's flight dynamics, which dictate the torque applied to the drum.

### 2.3.2 Proposed emulator topologies

The AWES test bench emulator simulates the interaction between the kite and the generator using an additional three-phase electrical machine (K), mechanically coupled to machine G. Machine K operates in opposition to G, acting as a motor during the reel-out phase to emulate the kite's mechanical forces, and as a generator during the reel-in phase to recover the mechanical energy applied by G. Two topologies are proposed in Fig. 3 for the emulator, each tailored to specific testing requirements.

The first topology, shown in Fig. 3b, uses two separate DC buses. Machine G connects to its own battery (G-B) through a DC-DC converter (G-C-DC) to store energy generated during the reel-out phase. Similarly, machine K is powered by a separate battery (K-B) through its own DC-DC converter (K-C-DC). This configuration closely replicates the energy storage and flow dynamics of a real AWES, making it ideal for studying energy storage requirements. However, it increases system complexity by requiring two batteries and two DC-DC converters.

The second topology, illustrated in Fig. 3c, simplifies the system by using a common DC bus shared by machines G and K. A single battery (J-B) and a single DC-DC converter (J-C-DC) manage energy storage. During the reel-out phase, energy generated by G is recirculated directly to K, reducing battery usage. This topology is more efficient and cost-effective, particularly for extended tests, but it sacrifices accuracy in emulating the distinct energy storage dynamics of a real AWES.





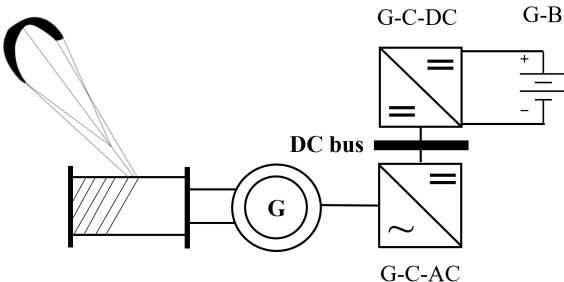

(a) Proposed power conversion topology for a real AWES.

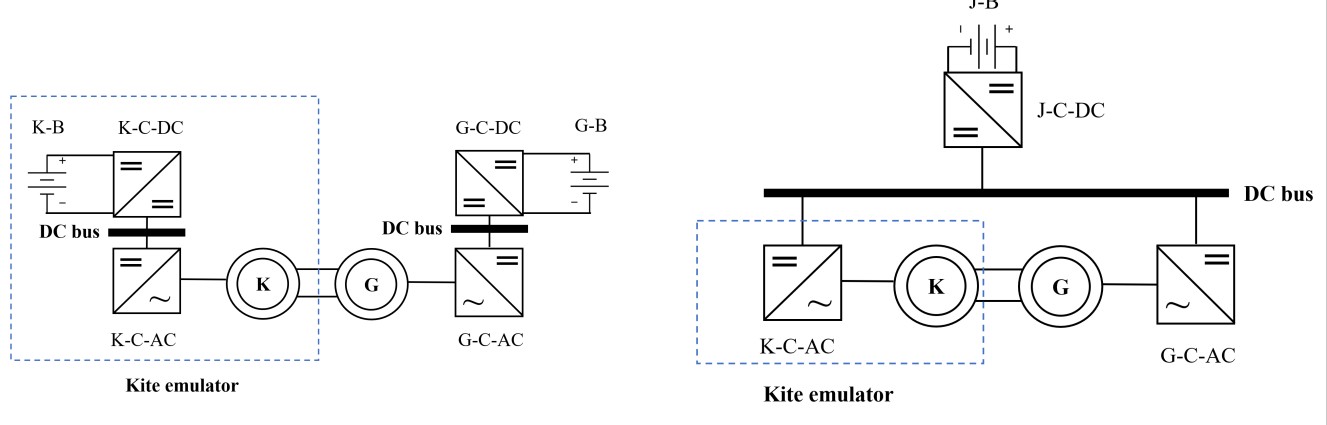

(b) Proposed two-DC bus topology approach for AWES electrical emulator.

(c) Proposed common DC bus topology approach for AWES electrical emulator.

**Figure 3.** Proposed power conversion topologies for the AWES test bench emulator.

## 2.4 Model predictive control strategy

The control scheme for the power converters is designed to address the dynamic and multi-variable nature of AWES. Compared to conventional wind energy systems, AWES are characterized by lower inertia, exposure to higher and more unpredictable wind conditions, and a highly variable flight cycle profile. These unique features demand a control strategy capable of rapid adjustments and robust performance under changing operating conditions.

MPC was selected for its capability to manage constraints, non-linearities, and fast-changing dynamics. Its excellent steady-state performance and rapid dynamic response make it particularly well-suited for AWES applications, where precise regulation of torque and energy flow is critical (Zhang et al., 2016; Hosseinzadeh et al., 2018). This section outlines the implementation of MPC for regulating the power converters in the proposed topologies (as shown in Sect. 2.3). Key features include a ripple



optimization strategy that minimizes fluctuations in controlled variables, ensuring that the kite emulator accurately reproduces the dynamic torque profile of a real AWES. Figure 4 illustrates the generalized structure of the MPC applied to a power converter, with specific details on its implementation for each system component provided in subsequent sections.

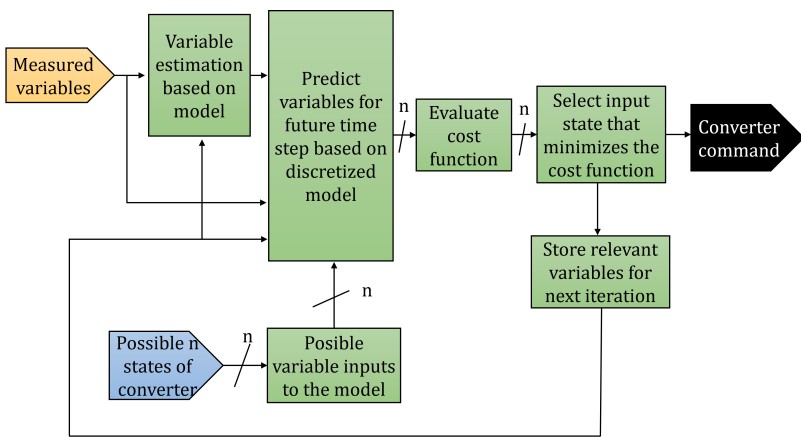

**Figure 4.** General MPC control scheme for power converters.

### 2.4.1 MPC ripple optimization vector strategy

The ripple optimization strategy is a critical enhancement of the proposed MPC, designed to minimize fluctuations in the controlled variables and improve the fidelity of the kite emulator. The MPC algorithm determines the switching states of either the bi-directional DC-DC converter or the two-level three-phase DC-AC converter. For the DC-DC converter, there are $n = 2$ possible states, while for the DC-AC converter $n = 7$ due to redundant states that produce the same voltage vector (Rodriguez et al., 2007). The selection of input states directly impacts the system's performance, particularly the accuracy of torque reference tracking and ripple suppression.

Two state selection strategies are evaluated:

- **Classical $S_i$ strategy:** This conventional approach selects a single optimal state $S_i$ (e.g., $S_0$ to $S_6$ for the DC-AC converter and $S_0$ to $S_1$ for the DC-DC converter) that minimizes the error between the predicted and reference output variables over the entire sampling period $T_s$. Once selected, this state remains constant throughout $T_s$. While computationally simple, this strategy can lead to higher ripple in the controlled variables, as it does not adjust for intermediate changes within $T_s$.

- **Symmetrical $S_i, S_0, S_i$ strategy:** This enhanced approach introduces a three-state symmetric input sequence $(S_i, S_0, S_i)$ to achieve finer control of the output variables. The controller calculates the fraction of $T_s$ during which $S_i$ will be applied




using a discrete parameter $f_{\text{duty}}$, which divides $T_s$ into three intervals:

$$T_1 = T_3 = \frac{1 - f_{\text{duty}}}{2} T_s, \quad (4)$$

$$T_2 = f_{\text{duty}} \cdot T_s. \quad (5)$$

Predefined discrete values for $f_{\text{duty}}$ are $\{0, 0.2, 0.4, 0.6, 0.8, 1\}$. Unlike the classical strategy, this method predicts the output variables at three points within $T_s$ ($T_1$, $T_2$, and $T_3$), enabling a closer match to the reference values.

The symmetrical $S_i, S_0, S_i$ strategy provides notable benefits, including reduced ripple in controlled variables, enhanced torque precision, and lower switching frequency. These advantages stem from the inclusion of the intermediate state $S_0$, which smooths transitions between $S_i$ states, as shown in Fig. 5. While this approach incurs a slightly higher computational cost, it is well-supported by modern microcontrollers (Yu and Long, 2024).

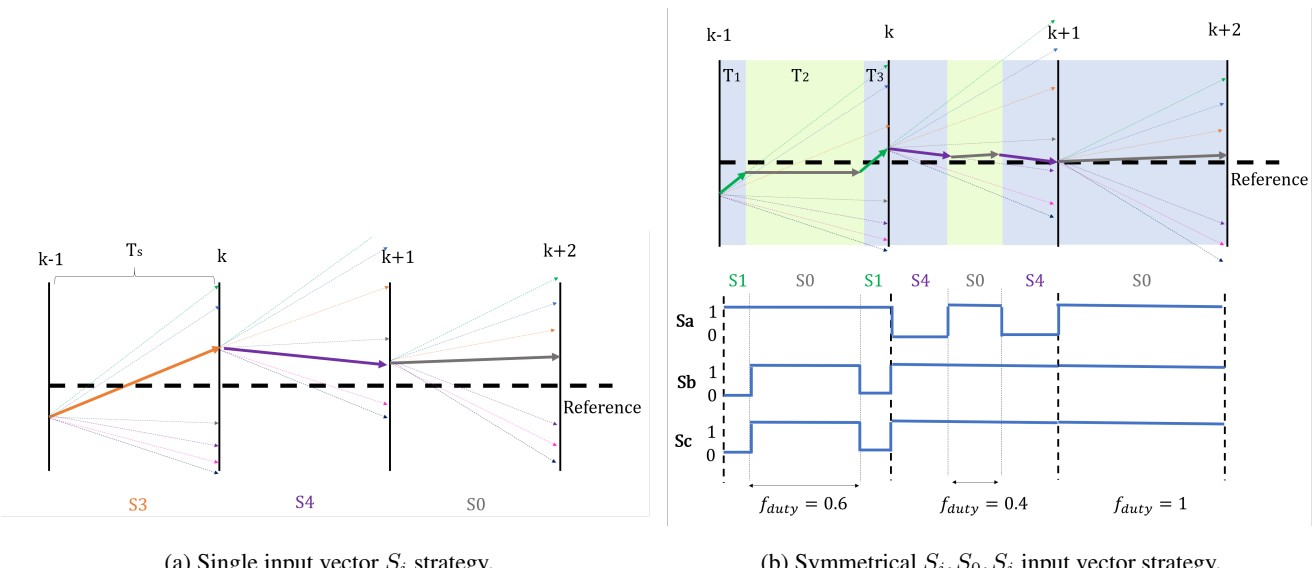

(a) Single input vector $S_i$ strategy.

(b) Symmetrical $S_i, S_0, S_i$ input vector strategy.

**Figure 5.** Comparison of MPC input vector strategies for three sample periods.

The comparison in Fig. 5 highlights the superior performance of the $S_i, S_0, S_i$ strategy in suppressing ripple and achieving greater accuracy over multiple sampling periods. This optimized strategy plays a vital role in ensuring that the kite emulator accurately replicates the dynamic torque profile of an actual AWES.

### 2.4.2 Model predictive control discretization

The discretization of the system models is essential for implementing MPC. Each component of the system is described in terms of discrete-time equations, enabling the MPC to predict and optimize the control variables. The number of switching states ($n$), and consequently the number of possible input variables for the model (as shown in Fig. 4), is determined by





the type of converter being controlled: $n = 7$ for a two-level three-phase DC-AC converter, which maps to seven distinct stator voltage vectors $V_s$, and $n = 2$ for a bi-directional DC-DC converter with two switching states. This section details the discretization process for the induction machine (IM), permanent magnet synchronous generator (PMSG), and bi-directional DC-DC converter.

**Induction Machine**

The induction machine's electrical model is expressed in matrix form as follows:

$$\mathbf{v} = \mathbf{A} \cdot \mathbf{s} + \mathbf{B} \cdot \mathbf{r} + \mathbf{C} \cdot \frac{d}{dt}\mathbf{s} + \mathbf{D} \cdot \frac{d}{dt}\mathbf{r}, \tag{6}$$

$$\mathbf{r} = \mathbf{E} \cdot \mathbf{s}. \tag{7}$$

Here, $\mathbf{v}$, $\mathbf{s}$, and $\mathbf{r}$ are vectors containing the system variables, including space vectors for the stator voltage $\overline{v_s}$, the stator flux $\overline{\varphi_s}$, stator current $\overline{i_s}$, rotor flux $\overline{\varphi_r}$, and rotor current $\overline{i_r}$, as shown in Eq. (8).

The values for matrices $\mathbf{A}$, $\mathbf{B}$, $\mathbf{C}$, and $\mathbf{D}$ are provided in Appendix A.

$$\mathbf{v} = \begin{bmatrix} \overline{v_s} \\ 0 \end{bmatrix}, \quad \mathbf{s} = \begin{bmatrix} \overline{\varphi_s} \\ \overline{i_s} \end{bmatrix}, \quad \mathbf{r} = \begin{bmatrix} \overline{\varphi_r} \\ \overline{i_r} \end{bmatrix}. \tag{8}$$

The electromagnetic torque $T_e$ is computed as

$$T_e = \frac{3}{2} p \, \mathrm{Im}\{\overline{\varphi_s}^* \cdot \overline{i_s}\}, \tag{9}$$

where $p$ is the number of pole pairs and $\mathrm{Im}\{...\}$ denotes the imaginary part of a complex number.

The model is discretized using the forward Euler method (Butcher, 2016):

$$\mathbf{s}_{k+1} = \mathbf{A}_v \mathbf{v}_k - \mathbf{A}_s \mathbf{s}_k. \tag{10}$$

The matrices $\mathbf{A}_v$ and $\mathbf{A}_s$ are calculated as

$$\mathbf{A}_v = (\mathbf{C} + \mathbf{D} \cdot \mathbf{E})^{-1} \cdot T_s, \tag{11}$$

$$\mathbf{A}_s = T_s \cdot (\mathbf{C} + \mathbf{D} \cdot \mathbf{E})^{-1} (\mathbf{A} + \mathbf{B} \cdot \mathbf{E}) - \mathbf{I}, \tag{12}$$

where $T_s$ is the sampling time, and $\mathbf{I}$ is the identity matrix.

**Permanent Magnet Synchronous Generator**

The PMSG is modeled in the synchronous reference frame as

$$\overline{v_s} = \mathbf{F} \cdot \overline{i_s} + \mathbf{G} \cdot \overline{\varphi_m} + \mathbf{L} \cdot \frac{d}{dt}\overline{i_s}, \tag{13}$$

where $\overline{v_s}$, $\overline{i_s}$, and $\overline{\varphi_m}$ are the space vectors for the stator voltage, stator current, and magnetic flux, respectively, defined as

$$\overline{v_s} = \begin{bmatrix} v_{sd} \\ v_{sq} \end{bmatrix}, \quad \overline{i_s} = \begin{bmatrix} i_{sd} \\ i_{sq} \end{bmatrix}, \quad \overline{\varphi_m} = \begin{bmatrix} \varphi_m \\ 0 \end{bmatrix}. \tag{14}$$





220     The values for matrices $\mathbf{F}$, $\mathbf{G}$, and $\mathbf{L}$ are provided in Appendix B.

The forward Euler method is applied to discretize the current space vector:

$$\overline{i}_{s\,k+1} = \mathbf{A}_u \overline{v}_{s\,k} + \mathbf{A}_i \overline{i}_{s\,k} - \mathbf{A}_{\varphi_m} \overline{\varphi_m}_{k}, \tag{15}$$

where the matrices are:

$$\mathbf{A}_u = T_s \cdot \mathbf{L}^{-1}, \tag{16}$$

225     $$\mathbf{A}_i = \mathbf{I} - T_s \cdot \mathbf{L}^{-1} \cdot \mathbf{F}, \tag{17}$$

$$\mathbf{A}_m = -T_s \cdot \mathbf{L}^{-1} \cdot \mathbf{G}. \tag{18}$$

The electromagnetic torque is calculated as

$$T_e = \frac{3}{2} p \left( i_{sd} \cdot i_{sq} \cdot (L_d - L_q) + \varphi_m \cdot i_{sq} \right), \tag{19}$$

where $L_d$ and $L_q$ are the d-axis and q-axis inductances, and $\varphi_m$ is the permanent magnet flux linkage.

230     **Bi-directional DC-DC converter**

The bi-directional DC-DC converter is modeled as

$$\mathbf{S} \cdot \mathbf{g} = \mathbf{I} \cdot \mathbf{m} + \mathbf{H} \cdot \frac{d}{dt} \mathbf{g}, \tag{20}$$

where $\mathbf{g}$ and $\mathbf{m}$ are vectors defined as

$$\mathbf{g} = \begin{bmatrix} V_{dc} \\ i_L \end{bmatrix}, \quad \mathbf{m} = \begin{bmatrix} V_{bat} \\ I_{dc} \end{bmatrix}. \tag{21}$$

235     Here, $V_{dc}$ is the DC bus voltage, $i_L$ is the inductor current of the converter, $V_{bat}$ is the battery voltage, and $I_{dc}$ is the DC bus current.

The matrices $\mathbf{S}$ and $\mathbf{H}$ are defined as.

$$\mathbf{S} = \begin{bmatrix} s_1 & 0 \\ 0 & s_1 \end{bmatrix}, \quad \mathbf{H} = \begin{bmatrix} 0 & L \\ -C_1 & 0 \end{bmatrix}, \tag{22}$$

where the variable $s_1$ represents the state of the bi-directional DC-DC converter's top switch (either 1 or 0), and the number
240     of possible states, $n$, in this case, is 2. The parameter $C_1$ represents the capacitance on the DC bus side of the converter, while
$L$ corresponds to the converter's inductance.

Discretizing with the forward Euler method gives:

$$\mathbf{g}_{k+1} = \mathbf{A}_m \mathbf{m}_k - \mathbf{A}_g \mathbf{g}_k, \tag{23}$$

where $\mathbf{A}_m$ and $\mathbf{A}_g$ are the system matrices





$$\mathbf{A}_g = T_s \cdot \mathbf{H}^{-1} \cdot \mathbf{S} + \mathbf{I}, \tag{24}$$

$$\mathbf{A}_m = T_s \cdot \mathbf{H}^{-1}. \tag{25}$$

The discretization simplifies the control implementation while maintaining the model's dynamic accuracy.

### 2.4.3 Control of the generating machine DC-AC converter (G-C-AC)

The G-C-AC converter ensures the electrical machine tracks the commanded rotational speed regardless of load torque. A proportional-integral (PI) controller calculates the reference electromagnetic torque, $T_{e_\text{ref}}$, from the mechanical speed error, which is then used by the model predictive control (MPC) as a reference.

For the induction machine (IM), the MPC employs the discretized model seen in Sect. 2.4.2 and minimizes the normalized errors of the stator flux and torque, as shown in the cost function:

$$\mathcal{J} = \lambda_\varphi \cdot \frac{|\|\overline{\varphi}_s\| - \varphi_{s_\text{ref}}|}{\varphi_{s_\text{base}}} + \lambda_T \cdot \frac{|T_e - T_{e_\text{ref}}|}{T_{e_\text{base}}}, \tag{26}$$

where $\lambda_\varphi$ and $\lambda_T$ are weighting factors and $\varphi_{s_\text{base}}$ and $T_{e_\text{base}}$ are base values used for normalization.

For the permanent magnet synchronous generator (PMSG), the MPC focuses on minimizing the torque error and reducing the $d$-axis stator current to enhance efficiency:

$$\mathcal{J} = \lambda_i \cdot \frac{|i_{sd}|}{i_{s_\text{base}}} + \lambda_T \cdot \frac{|T_e - T_{e_\text{ref}}|}{T_{e_\text{base}}}. \tag{27}$$

where $\lambda_i$ and $\lambda_T$ are weighting factors and $i_{s_\text{base}}$ and $T_{e_\text{base}}$ are base values used for normalization.

Control diagrams for both machine types are shown in Fig. 6.

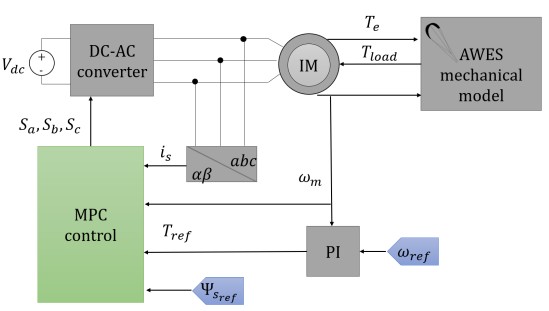
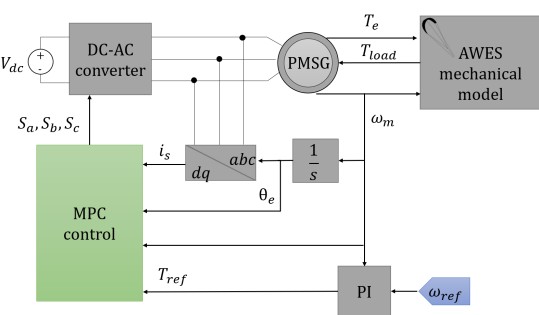

(a) Control scheme for the generator machine DC-AC converter (G-C-AC) when an induction machine is used.

(b) Control scheme for the generator machine DC-AC converter (G-C-AC) when a permanent magnet synchronous generator is used.

**Figure 6.** Control schemes for the generator machine DC-AC converter (G-C-AC).





### 2.4.4 Control of the kite emulator machine DC-AC converter (K-C-AC)

The K-C-AC converter applies a reference torque to emulate the kite's mechanical behavior during the AWES cycle. This torque is computed using the single-mass mechanical model:

$$J_{\text{eq}} \cdot \frac{d\omega_m}{dt} = T_e - T_{\text{load}}, \tag{28}$$

where $J_{\text{eq}}$ is the equivalent inertia, $T_e$ is the electromagnetic torque, and $T_{\text{load}}$ is the load torque.

Similar to the G-C-AC, the MPC for the K-C-AC uses the machine's discretized model and minimizes reference variable errors acording to Equation 26 for the IM and Eq. (27) for the PMSG. Control schemes for the IM and PMSG implementations are presented in Fig. 7.

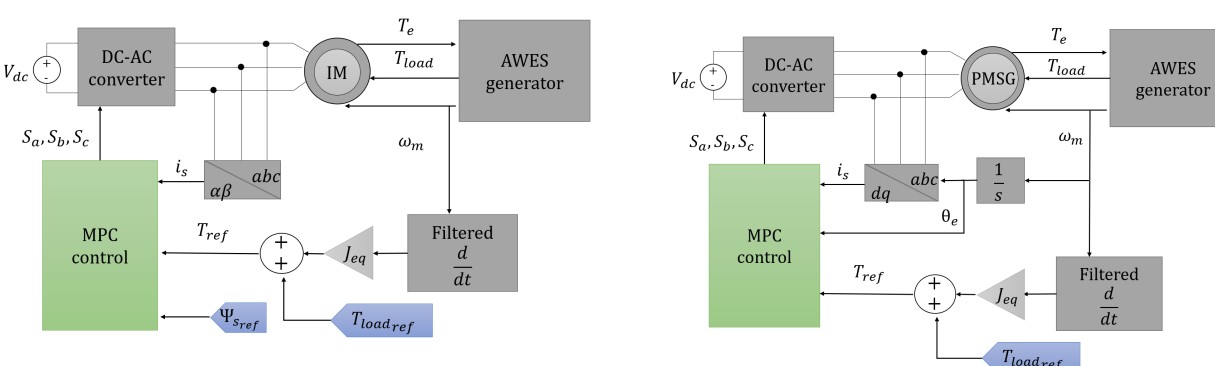

(a) Control scheme for the emulator machine DC-AC converter (K-C-AC) when an induction machine is used.

(b) Control scheme for the emulator machine DC-AC converter (K-C-AC) when a permanent magnet synchronous generator is used.

**Figure 7.** Control schemes for the emulator machine DC-AC converter (K-C-AC).

### 2.4.5 Control of the DC-DC converter

The DC-DC converter maintains a constant DC bus voltage by regulating power flow between the battery and the DC bus. The MPC uses measured currents and voltages to minimize the normalized errors in battery current ($i_L$) and DC bus voltage ($V_{\text{dc}}$) using the cost function:

$$\mathcal{J} = \lambda_{i_L} \cdot \frac{|i_L - i_{L_{\text{ref}}}|}{i_{L_{\text{base}}}} + \lambda_v \cdot \frac{|V_{\text{dc}} - V_{\text{dc}_{\text{ref}}}|}{V_{\text{dc}_{\text{base}}}}, \tag{29}$$

where $\lambda_{i_L}$ and $\lambda_v$ are weighting factors and $V_{\text{dc}_{\text{base}}}$ and $i_{L_{\text{base}}}$ are base values used for normalization. The control scheme for the
DC-DC converter maintaining a stable DC bus is found in Fig. 8





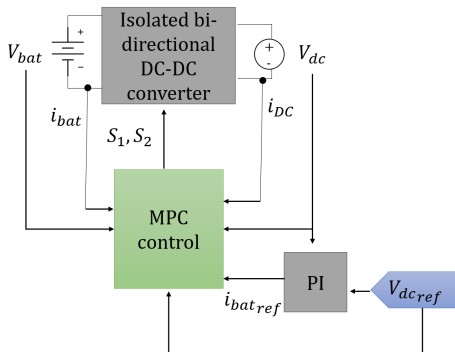

**Figure 8.** Control scheme for the DC-DC converter to maintain a steady DC bus voltage.

## 3 Proposed case study

To validate the AWES electric emulator topology and the proposed MPC control, real experimental flight data from a commercial AWES device, following an optimal mechanical path at an average wind speed of $7\,\mathrm{m\,s^{-1}}$, were used. These results validate the proposed emulator for AWES applications of similar characteristics. The recorded flight time series data for two cycles, including tether tension and tether length variation, were obtained from Schmehl (2023). These linear variables were translated into torque and rotational speed references for the emulator using the methodology in Sect. 2.2, assuming a drum radius of $R_{\mathrm{drum}} = 0.3\,\mathrm{m}$.

### 3.1 Case study and test environment

Experimental mechanical profiles from Schmehl (2023), representing two AWES cycles, were first tested on a real AWES generator (Fig. 3a) using the proposed MPC control for two electrical machines: an induction machine (IM) and a permanent magnet synchronous generator (PMSG). Subsequently, the proposed AWES electrical test bench emulator topologies (Figs. 3b and 3c) were tested under the same control strategy and experimental data. Both IM and PMSG machines were considered for the test bench emulator.

The tests were conducted in the MATLAB/Simulink environment, with all electrical hardware modeled using Simscape Electrical. Parameters for modeling and control are detailed in Appendix C.

### 3.2 Control objectives and performance indicators

The main objectives of the proposed MPC control, along with their corresponding performance metrics, are as follows:

– **Accurate torque tracking with minimal ripple:** Ensure that the kite emulator machine precisely follows the reference load torque of the AWES while minimizing electromagnetic torque ripple. Performance is evaluated using the normalized





root mean squared error (RMSE) between the measured load torque on the shaft and the reference torque from the AWES
dynamic profile.

    – **Precise speed regulation of the generator machine:** Maintain accurate tracking of the optimal reference speed during
both transient and steady-state operation. This is assessed using the normalized RMSE between the measured mechanical
shaft speed and the reference speed from the AWES dynamic profile.

– **Maximization of energy conversion efficiency:** Optimize the total electrical energy extracted from the available me-
chanical energy of the AWES cycle. Efficiency is quantified as the ratio of total energy stored in the battery to the total
mechanical energy generated by the kite.

    – **Stable and regulated DC bus voltage:** Ensure a steady and well-regulated DC bus voltage throughout operation. This
is evaluated using the normalized RMSE between the measured DC bus voltage and the reference voltage.

## 4   Results

This section presents the performance evaluation of the proposed AWES test bench emulator, highlighting its ability to replicate
the dynamic behavior of a real AWES. Key numerical performance metrics are summarized in Table 1. Experimental results
for the kite emulator and generator dynamics are detailed in Fig. 9, focusing on the torque, speed, and power profiles for two
optimal figure-8 AWES cycles.

### 4.1   Numerical performance metrics

Table 1 compares the overall and maximum energy efficiencies ($\eta_{\text{total}}$ and $\eta_{\text{max}}$) for PMSG and IM machines, along with root
mean square errors (RMSE) for speed, torque, and DC bus voltage. The table also includes battery performance metrics, such as
the number of AWES cycles required to charge or discharge the batteries for both separated and common DC bus topologies.
PMSG machines show slightly higher efficiency, particularly during reel-in phases, while IM machines exhibit comparable
accuracy in torque and speed tracking.

### 4.2   Emulated mechanical dynamics

Figure 9 showcases the performance of the AWES emulator for two figure-8 cycles using PMSG machines for both the genera-
tor and kite emulator. Figure 9a highlights the torque applied by the emulator, accurately replicating the reference torque from
the real AWES with low ripple and high precision. Similarly, Fig. 9b demonstrates the generator's ability to track the optimal
rotational speed with less than 1% RMSE. These results correspond to the case with separated DC buses; however, the torque
and rotational speed profiles for the common DC bus configuration were nearly identical, indicating that the bus topology has
minimal impact on these variables.





**Table 1.** Performance evaluation of AWES and test bench emulator across generator types and topologies.

| AWES | | | | | AWES test bench emulator | | | | | | | |
|---|---|---|---|---|---|---|---|---|---|---|---|---|
| Generator | $\eta_{\text{total}}$ (pu) | $\eta_{\text{max}}$ (pu) | AWES cycles to charge G-B | Speed RMSE (%) | Emulator | Battery topology | Torque RMSE (%) | Speed RMSE (%) | $V_{dc}$ RMSE (%) | AWES cycles to charge G-B | AWES cycles to discharge K-B | AWES cycles to discharge J-B |
| PMSG | 0.82 | 0.93 | 557 | 0.6 | PMSG | Separated | 0.44 | 0.6 | 0.11 | 563 | 390 | - |
| | | | | | | Common | 0.43 | 0.6 | 0.15 | - | - | 1460 |
| | | | | | IM | Separated | 0.96 | 0.48 | 0.11 | 569 | 382 | - |
| | | | | | | Common | 0.94 | 0.48 | 0.14 | - | - | 1188 |
| IM | 0.78 | 0.91 | 586 | 0.04 | IM | Separated | 1.01 | 0.04 | 0.1 | 596 | 382 | - |
| | | | | | | Common | 0.97 | 0.05 | 0.13 | - | - | 1087 |

### 4.3 Battery performance, power profiles and efficiency analysis

The filtered power profiles in Fig. 9c compare the energetic performance of separated and common DC bus configurations.
The separated configuration shows distinct charging and discharging profiles for G-B and K-B, resembling the behavior of real
AWES batteries. The common DC bus configuration achieves significant power recirculation, reducing reliance on the battery
and enabling 280–370% more AWES cycles per charge. However, the separated topology is better suited for storage-focused
studies.

Figure 10 provides the instantaneous efficiency profiles of IM and PMSG machines during the generating (reel-out) phase
of one AWES cycle. PMSG consistently outperforms IM at low power levels, with a maximum efficiency difference of 6%. At
higher power levels, both machines achieve similar efficiency, with differences below 2%.

## 5 Conclusions

This paper presents a validated electric topology and torque ripple-optimizing model predictive control (MPC) strategy for an
airborne wind energy system (AWES) generator and its corresponding test bench electrical emulator. While extensive research
has been conducted on optimizing AWES flight trajectories and aerodynamic performance, the mechanical-to-electrical power
conversion process remains underexplored. This study addresses that gap by reviewing key AWES power conversion architec-
tures and validating an efficient control framework that accurately emulates AWES energy dynamics using experimental data
from two AWES flight cycles.

The proposed electric topology and MPC control effectively convert the mechanical energy extracted from an optimal AWES
flight path into electrical energy, achieving a total system efficiency exceeding 80% across two complete flight cycles (reel-in
and reel-out). The system maintains precise speed control, with the generator tracking the reference rotational speed within a
1% root-mean-square error (RMSE). Both permanent magnet synchronous generators (PMSG) and induction machines (IM)



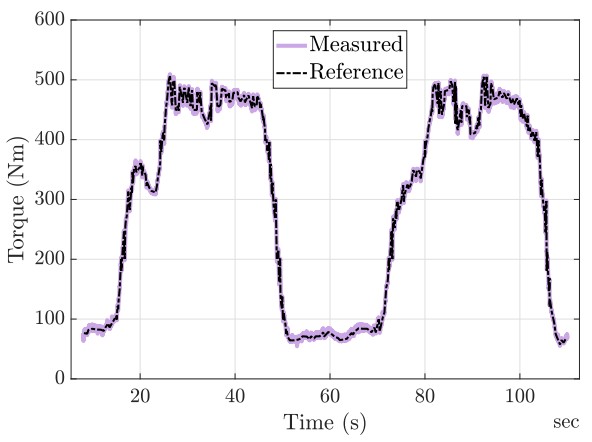

(a) Emulated kite torque.

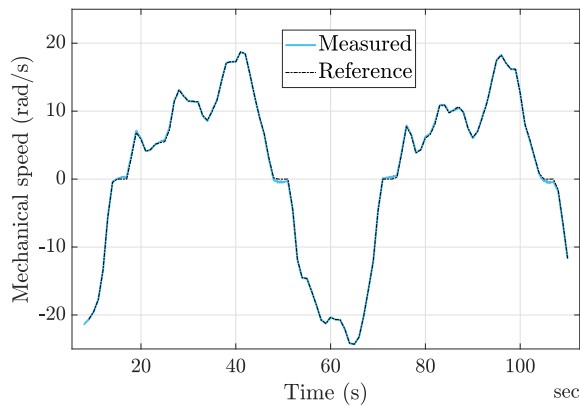

(b) Mechanical speed of the system.

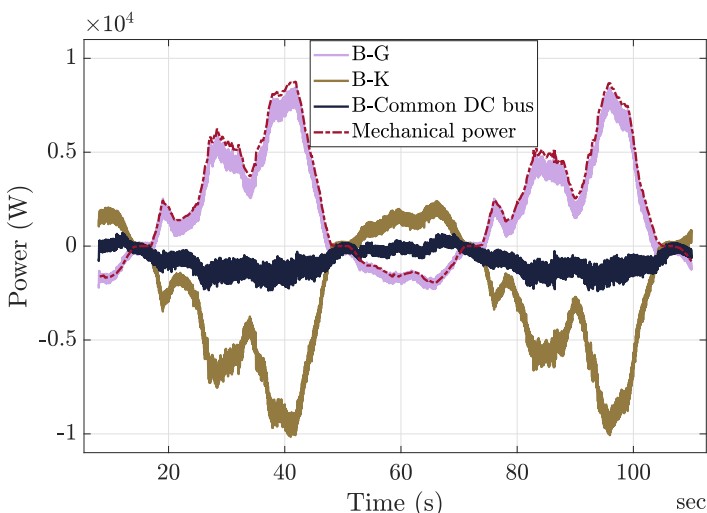

(c) Filtered power readings for the generator and emulator batteries.

**Figure 9.** Case study results for two optimal figure-8 cycles by the AWES emulator using PMSG machines.

were evaluated as potential AWES generator options, with PMSG machines demonstrating superior performance, achieving 4% higher total energy efficiency and instantaneous efficiency improvements of 2–6% compared to IM machines. Both machines exhibit increasing efficiency at higher instantaneous power levels, with peak efficiencies reaching 93%, underscoring the importance of integrating electrical efficiency considerations into AWES flight path design.

The test bench emulator presented in this paper accurately reproduces the mechanical dynamics of an AWES, providing a controlled environment for evaluating power conversion and control strategies. The proposed torque ripple-optimized MPC ensures that the emulator machine follows the reference torque profile with an RMSE below 1%, effectively simulating the




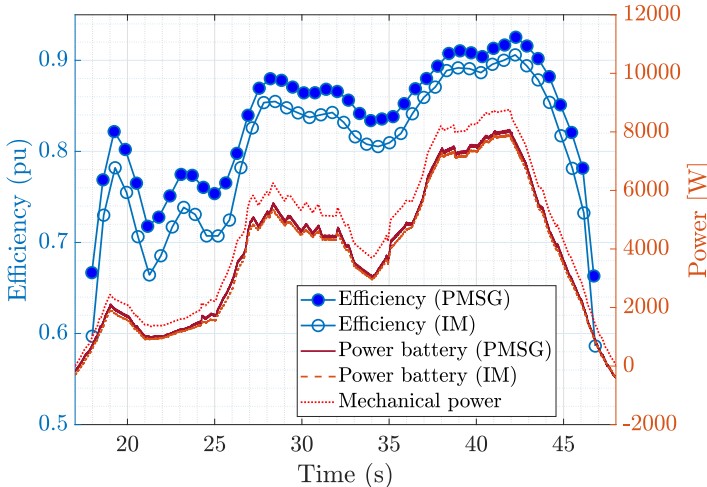

**Figure 10.** Instantaneous energy efficiency of the power conversion system during the generating phase of one AWES cycle.

mechanical forces experienced by an AWES generator. While both PMSG and IM machines perform effectively in the emulator, IM machines exhibit slightly higher torque tracking errors (0.5%) and a 2% reduction in energy efficiency compared to PMSG machines. These results highlight the flexibility of the test bench to accommodate different machine types while maintaining precise control and energy conversion.

        Two test bench topologies were analyzed: a separated DC bus topology, which closely mimics the energy storage behavior
of real AWES ground stations, and a common DC bus topology, which significantly reduces battery requirements by facilitating direct energy recirculation between the generator and emulator machines. The separated DC bus topology achieves 98% accuracy in replicating real AWES energy storage dynamics but requires 45–55% greater battery capacity for the emulator. The common DC bus topology, while less suited for energy storage studies, is highly effective for long-duration control tests, reducing battery dependency to just one-third of the separated configuration.

The findings in this study highlight the effectiveness and versatility of the proposed AWES power conversion strategies and emulator. By filling a gap in AWES research, this work provides a foundation for integrating mechanical and electrical efficiency considerations into AWES system design, making it highly relevant not only for power system researchers but also for those specializing in AWES aerodynamics and flight control. Future work should focus on co-optimizing flight trajectories and power conversion strategies, exploring efficiency gains at varying power levels, and expanding the test bench framework
to support grid integration and larger-scale AWES applications.

*Author contributions.* Conceptualization, D.S. and C.N; methodology, C.N., D.S, F.D. and J.G.; software, C.N..; validation, C.N.; formal analysis, C.N and F.D.; investigation, C.N.; resources, D.S..; data curation, C.N and J.G..; writing—original draft preparation, C.N. and F.D.;



writing—review and editing, C.N., F.D., D.S. and J.G.; visualization, C.N., D.S, F.D. and J.G; supervision, D.S.; project administration, D.S. All authors have read and agreed to the published version of the manuscript.

*Competing interests.* The authors declare no competing interests.

*Acknowledgements.* This work is part of the project PID2022-141520OB-I00 funded by MICIU/AEI/ 10.13039/501100011033. Work by F.D.-N. was supported by the grant with reference IND2022/AMB-23521 funded by Comunidad de Madrid. AI tools have been used exclusively to proofread and format specific parts of the manuscript.





**Appendix A: Values for induction motor model matrices using a static reference frame**

The following matrices define the state-space representation of the induction motor (IM) model in a stationary reference frame. In these matrices, $R_s$ and $R_r$ represent the stator and rotor resistances, respectively, while $L_s$, $L_r$, and $L_m$ denote the stator, rotor, and magnetizing inductances. The parameter $\omega_e$ corresponds to the electrical angular speed of the rotor.

$$\mathbf{A} = \begin{bmatrix} 0 & R_s \\ 0 & 0 \end{bmatrix}, \qquad \mathbf{B} = \begin{bmatrix} 0 & 0 \\ -j\omega_e & R_r \end{bmatrix}, \qquad \mathbf{C} = \begin{bmatrix} 1 & 0 \\ 0 & 0 \end{bmatrix},$$

$$\mathbf{D} = \begin{bmatrix} 0 & 0 \\ 1 & 0 \end{bmatrix}, \qquad \mathbf{E} = \begin{bmatrix} \frac{L_r}{L_m} & \frac{L_m^2 - L_r L_s}{L_m} \\ \frac{1}{L_m} & -\frac{L_s}{L_m} \end{bmatrix}. \qquad (A1)$$

**Appendix B: Values for the PMSG model matrices using a dq reference frame**

The following matrices describe the state-space representation of the Permanent Magnet Synchronous Generator (PMSG) model in a rotating $dq$ reference frame. Here, $R_s$ represents the stator resistance, while $L_d$ and $L_q$ correspond to the direct-axis and quadrature-axis inductances. The parameter $\omega_e$ denotes the electrical angular speed of the rotor.

$$\mathbf{F} = \begin{bmatrix} R_s & -L_d\omega_e \\ L_q\omega_e & R_s \end{bmatrix}, \qquad \mathbf{G} = \begin{bmatrix} 0 & -\omega_e \\ \omega_e & 0 \end{bmatrix}, \qquad \mathbf{L} = \begin{bmatrix} L_d & 0 \\ 0 & L_q \end{bmatrix}. \qquad (B1)$$

**Appendix C: Case Study Parameters**

This section summarizes the key electrical and control parameters used in the case study, including battery specifications, converter characteristics, and machine properties. These values define the operational limits and dynamic behavior of the AWES test bench emulator.

| Battery Parameter | Value |
|---|---|
| Nominal voltage ($V_{nom}$) | 360 V |
| Internal resistance ($R_{bat}$) | 0.1 Ω |
| Usable energy | 14.76 kWh |

**Table C1.** Battery parameters.





| DC-DC Converter Parameter | Value |
|---|---|
| Capacitance 1 ($C_1$) | $4.7 \cdot 10^{-4}$ F |
| Capacitance 2 ($C_2$) | $1 \cdot 10^{-4}$ F |
| Inductance ($L$) | $1.4 \cdot 10^{-3}$ H |
| Resistance 1 ($R_1$) | $1 \cdot 10^{-3}$ $\Omega$ |
| Resistance 2 ($R_2$) | $1 \cdot 10^{-3}$ $\Omega$ |
| Switching frequency ($f_{switch}$) | 10 kHz |

**Table C2.** DC-DC converter parameters.

| IM Parameters | | PMSG Parameters | |
|---|---|---|---|
| Moment of inertia ($J_{eq}$) | $2.5\,\mathrm{kg \cdot m^2}$ | Moment of inertia ($J_{eq}$) | $2.72\,\mathrm{kg \cdot m^2}$ |
| Pole pairs ($p$) | 8 | Pole pairs ($p$) | 8 |
| Leakage inductance ($L_{lr}$) | 0.1931 pu | $d/q$ inductance ($L_d, L_q$) | $15 \cdot 10^{-3}$ H |
| Leakage inductance ($L_{ls}$) | 0.1316 pu | Flux linkage ($\varphi_m$) | 0.85 Wb |
| Mutual inductance ($L_m$) | 5.3833 pu | Stator resistance ($R_s$) | 0.2 $\Omega$ |
| Rotor resistance ($R_r$) | 0.0658 pu | Nominal power ($P_n$) | 20 kW |
| Stator resistance ($R_s$) | 0.0302 pu | Nominal voltage ($V_n$) | 380 V |
| Nominal power ($P_n$) | 20 kW | | |
| Nominal voltage ($V_n$) | 380 V | | |

**Table C3.** Parameters for induction machine (IM) and permanent magnet synchronous generator (PMSG).

| Control DC-DC Parameters | | Control DC-AC Parameters | |
|---|---|---|---|
| Integral gain ($K_i$) | 45 | Integral gain ($K_i$) | 200 |
| Proportional gain ($K_p$) | 1 | Proportional gain ($K_p$) | 250 |
| Cost weight ($\lambda_{i_L}$) | 1 | Cost weights | $\lambda_i = 1, \lambda_T = 1$ |
| Switching frequency ($f_{switch}$) | 10 kHz | | $\lambda_\varphi = 2, \lambda_T = 1$ |

**Table C4.** Control parameters for DC-DC and DC-AC converters.



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
