# Peer review of "Airborne wind energy system test bench electrical emulator"

_Wind Energy Science, 2025_

## Author Response (AR1)

1) Referee comment: The authors have referred to the state of the art. However, the introduction could be strengthened by explicitly highlighting how this work significantly advances the field beyond existing emulators and control strategies. (Emphasizing the specific limitations of previous emulators, mention unique benefits of the proposed approach). Perhaps a sub-section in the introduction which addresses the novelty, scope and limitations of the presented work.

Authors' response: Agreed. The introduction has been strengthened by adding a dedicated subsection "Novelty, scope and limitations" (lines 60-93), which explicitly highlights how the proposed work advances beyond existing emulators and control strategies, emphasizing the specific limitations of previous approaches and the unique benefits of the emulator presented.

2) Referee comment: Furthermore on scope, it will be helpful in knowing what part was modelled, what part was measured from experiments, and what information was taken from another source.

Authors' response: Agreed. The new "Novelty, scope and limitations" (lines 90-104) section within the introduction clarifies that the machines, converters, storage, and control are modelled; torque and speed profiles come from experimental AWES flight data; and remaining parameters are based on values from literature and reference texts adapted to this study.

3) Referee comment: The authors chose PMSG and IM for the case study, are these and the parameters for the emulator representative of the state of the art AWES prototypes?

Authors' response: We thank the reviewer for this observation. The PMSG and IM parameters were taken from well-documented 10–30 kW wind-energy machines, consistent with those used in early AWES prototypes (e.g., Ampyx 12 kW, TU Delft 20 kW). We acknowledge that ideally AWES-specific electrical data would be used; however, to the best of our knowledge such data are not yet widely available. A note has been added in Appendix C (lines 485–493) to clarify this and to indicate that the methodology can be updated as such data become available.

4) Referee comment: The AWES schematic shown in Fig.1 is specific to the presented application? Maybe first define or illustrate the state of the AWES systems and its main components and operational framework and then show the presented AWES system and emulator.

Authors' response: We agree that providing an overview of the state of AWES systems and their main components before introducing our specific application improves clarity. In the revised manuscript, we have added a paragraph in Section 2.1 that illustrates the state of the art of electrical topologies used in AWES systems and explains their main components and operational framework (see lines 123–135).

We also clarify that (now) Fig. 2 is intended as a generic diagram, since details such as the specific energy storage technology or power converters can vary between prototypes. This generic topology reflects most of the available information (which remains scarce, to the best of our knowledge) on the electrical conversion systems of ground-based AWES.

Finally, in Section 2.3.1 we now explain in greater detail how the specific topology used in our case studies fits within this presented generic framework (see lines 169–173).

5) Referee comment: In my opinion, adding a flow chart in the methodology section showing the overall procedure and work flow of the presented work will be of great help. Example showing the models that were developed, the experimental data, load estimation, validation cases, results, error comparison.... Main idea is for the reader to get a complete understanding of what was done, and how was it done without reading the text.

Authors' response: We fully agree that a flowchart improves clarity and provides a quick overview of the workflow. Accordingly, the revised manuscript now includes the recommended flowchart summarizing the overall procedure (see new Figure 1 in the Methodology section).

6) Referee comment: The experimental validation details need more detail.

The authors only gave reference to a previous study from where the flight time series and data were obtained. More detail on the experimental setup should be provided.

Authors' response: Agreed. The revised version of the manuscript now includes a new table (Table 1) summarizing the most relevant details of the datasets used for the validation of the proposed emulator. In addition, as suggested by Reviewer 2, we have incorporated a second dataset for validation to further strengthen the experimental section, which is also included in the table.

7) Referee comment: In the discussion or conclusion section, the authors are requested to highlight limitations of the presented emulator and control strategy. For example, how do the assumptions taken for estimating the torque loads hold in real-world application.

Authors' response: The revised conclusions now highlight limitations of the presented emulator and control strategy, noting the need for experimental validation and the use of more detailed or ground-station-specific mechanical models to better estimate rotational variables and reflect real-world torque loads, among others (lines 455-461).

8) Referee comment: Minor remarks regarding acronyms, they should be defined when first used. Some acronyms were defined multiple times (example RMSE and PMSG).

Authors' response: This has been revised and corrected throughout the new version of the manuscript

**Reviewer 2: RC2**

1) Referee comment: Controller Benchmarking:

The authors justify the selection of model predictive control (MPC) by referencing the fast-changing dynamics of AWE systems. While the MPC performance appears promising, it is essential to include a comparative analysis with at least one other control strategy, such as a conventional

PID or another baseline controller, to validate the superiority of MPC in this context.

Authors' response: We thank the reviewer for this valuable comment. We have further strengthened the justification for selecting MPC by citing additional benchmarking evidence from the literature, demonstrating its superior dynamic performance and efficiency compared to conventional strategies such as FOC and DTC in related high-performance drive applications. This revised justification has been included in Section 2.4 of the manuscript.

We agree that, while existing evidence strongly supports MPC's fast dynamic response capabilities (see lines 199-210), a direct experimental comparison with alternative control strategies (e.g., PID or FOC) under AWES-specific conditions would further validate its superiority. Conducting such benchmarking, however, is beyond the scope of this paper and is now explicitly noted as a limitation and an area for future work in lines 457-458 in the conclusion.

**2) Referee comment: Novelty of the Test Rig:**

The paper lacks a clear articulation of the novelty of the proposed test bench. The authors should elaborate on how their emulator differs from and improves upon existing AWE test platforms discussed in the literature.

Authors' response: Agreed. The revised manuscript now explicitly articulates the novelty of the proposed emulator. A dedicated subsection "Novelty, scope and limitations" at the end of the Introduction has been added, clearly explaining how this work advances beyond existing AWE test platforms by comparing two machine types under realistic AWES profiles, evaluating two distinct DC-bus configurations, and introducing an MPC strategy that actively reduces torque ripple.

**3) Referee comment: Validation with Real Data:**

Although the authors use real AWE test data to validate their numerical model, relying on a single data set may not be sufficient. Additional validation using at least one more independent AWE operation dataset would strengthen the credibility of the model's performance.

Authors' response: We agree with the reviewer's observation. In the revised manuscript, validation of the proposed topologies and control strategy has been extended to include an additional independent AWE dataset alongside the original one. The results for both datasets are now presented and discussed, providing stronger evidence of the model's credibility and robustness.

**4) Referee comment: Physical Implementation:**

The paper does not clarify why only a numerical/simulation model was developed and no physical test bench was built. The authors should justify this decision and discuss the implications for the practical application of their work.

Authors' response: Agreed. We have clarified why only a numerical/simulation model was developed. This choice was made as an initial design step, allowing systematic comparison of topologies and control strategies before hardware implementation. This justification and its implications for practical application are now included in the new "Novelty, scope and limitations" section at the end of the Introduction. Experimental validation is also identified as key next step of the presented work in lines 455-461 of the conclusions.

**5) Referee comment: **DC Bus Power Observation:**

In Figure 9, the power at the common DC bus is nearly zero throughout the operation. This suggests that most of the generated power is consumed internally for kite emulation, which significantly diverges from real-world AWE system performance. If grid integration is planned for future work, this could pose a challenge, as the generated power would need to be dispatched to the grid rather than being fully consumed by the emulator. The authors should clarify the rationale for selecting a common DC bus topology and discuss its practical implications.

Authors' response: Correct. One of the findings of this article is that the common DC bus topology is a test-bench mode that recirculates energy internally, greatly reducing battery stress and enabling many repeated control-strategy tests. However, it is not intended to emulate grid-tied

operation; for studies of realistic storage behavior or grid integration, the separated DC bus topology should be used.

This point was not sufficiently clear in our original version of the manuscript. The rationale and distinction are now explicitly stated in the revised manuscript in Section 1.1 (lines 77–84), Section 2.3.2 (lines 185–197), Section 4.3 (lines 399–420), and summarized again in the Conclusions (lines 442–451).

**6) Referee comment: Topological Comparison:**

A more detailed comparative analysis between the common DC bus and separate DC bus topologies is recommended. This would provide deeper insights into the trade-offs and performance implications of each configuration.

Authors' response: Agreed. Section 4.3 of the revised manuscript presents a more detailed comparative analysis of the common DC bus and separate DC bus topologies. Additionally, a new table (Table 4 in the revised manuscript) has been included to summarize the key insights for clarity.

**7) Referee comment: **Enhancement of Results Section:**

The results section would benefit from additional data and analysis that reflect the above concerns. This includes more extensive validation, clearer discussion of control performance, and practical considerations related to topology and physical implementation.

Authors' response: We agree. The revised manuscript includes an additional dataset and a more detailed Results section, providing broader validation, clearer discussion of control performance, and expanded analysis of topology and implementation aspects.